# Biochemical Characterization of Black and Green Mutant Elderberry during Fruit Ripening

**DOI:** 10.3390/plants12030504

**Published:** 2023-01-22

**Authors:** Maja Mikulic-Petkovsek, Anton Ivancic, Sasa Gacnik, Robert Veberic, Metka Hudina, Silvija Marinovic, Christian Molitor, Heidi Halbwirth

**Affiliations:** 1Chair for Fruit Growing, Viticulture and Vegetable Growing, Department of Agronomy, Biotechnical Faculty, University of Ljubljana, Jamnikarjeva 101, SI-1000 Ljubljana, Slovenia; 22 Chair for Genetics, Faculty of Agriculture and Life Sciences, University of Maribor, Pivola 10, SI-2311 Hoce, Slovenia; 3Institute of Chemical, Environmental and Bioscience Engineering, Technische Universität Wien, Getreidemarkt 9, A-1060 Vienna, Austria

**Keywords:** black elderberry, *Sambucus nigra*, green genotype, polyphenolics, sugars, organic acids, phenylpropanoid enzymes

## Abstract

The content of sugars, organic acids, phenolic compounds and selected enzyme activities in the anthocyanin pathway were analyzed in NIGRA (*Sambucus nigra* var. *nigra*—black fruits) and VIRIDIS (*S. nigra* var. *viridis*—green fruits) fruits over four stages of ripening. The share of glucose and fructose in green fruits was higher than in colored fruits, and the sugar content increased significantly until the third developmental stage. Ripe NIGRA berries had 47% flavonol glycosides, 34% anthocyanins, 3% hydroxycinnamic acids and 14% flavanols, whereas the major phenolic group in the VIRIDIS fruits, making up 88% of the total analyzed polyphenols, was flavonols. NIGRA fruits were rich in anthocyanins (6020 µg g^−1^ FW), showing strong activation of the late anthocyanin pathway (dihydroflavonol 4-reductase, anthocyanidin synthase). In both color types, phenylalanine ammonia lyase and chalcone synthase/chalcone isomerase activities were highest in the first stage and decreased during ripening. In VIRIDIS fruit, no anthocyanins and only one flavanol (procyanidin dimer) were found. This was most likely caused by a lack of induction of the late anthocyanin pathway in the last period of fruit ripening. The VIRIDIS genotype may be useful in studying the regulatory structures of anthocyanin biosynthesis and the contribution of distinct flavonoid classes to the health benefits of elderberries.

## 1. Introduction

*Sambucus nigra* L. (common elder/elderberry, black elder/elderberry, European elder/elderberry), which belongs to the genus *Sambucus* and the family Adoxaceae, is one of the most popular and most frequently studied medicinal plant species. The taxonomy of the genus *Sambucus* remains a debatable and not completely resolved puzzle. Bolli [1] reduced the number of species from more than 30 to only 9, but a study published by Applequist [2] suggested that the actual number of ‘true’ species is probably higher. Based on our 20 years of experience with crosses among different *Sambucus* species, we would surmise that there are probably between 12 and 20 species. Many of the elderberry species, if not all, have some useful medicinal properties. In addition, elderberry juice is a valuable food additive [3].

Black elderberries are deciduous, shrubby to arborescent herbs up to 10 m high. The bark is usually beige-gray, conspicuously dotted with orbicular, dark brown lenticels. The leaves are opposite and odd-pinnate, and 5–30 cm long. The leaflets are serrate, and their number can vary from 3 to 15, usually between 5 and 9. The inflorescences are compound, flat topped cymes with numerous creamy-white, rarely pink or light purplish-red, actinomorphic, pentamerous flowers that are approximately 5 mm in diameter and intensely fragrant. The fruits are small, berry-like drupes, 5–8 mm in diameter. In full maturity, their color is usually purplish-black, although there are also genotypes with a grayish-green or whitish green color, such as the botanical variety *viridis* [1,4].

*Sambucus nigra* var. *viridis* Weston (syn.: *S. nigra* var. *alba* Weston*, S. nigra f. alba* (Weston) Rehder) represents a selection of black elder characterized by whitish-green or yellow-greenish fruits. No reliable data could be found in the literature about its origin. The first scientific names of this variety originate from the 17th century. It was mentioned by Richard Weston (1733–1806) [5], meaning this is probably a very old variety. It probably appeared as a natural mutant, or as a result of random genetic segregation. If the second hypothesis is true, then it is probably multiple recessive (recessive homozygote in more than one gene locus). If it appeared as recessive in one locus, then such ‘light’ fruited genotypes would appear more frequently. It is not ‘albino’, because plants are normally green (in fact, the green pigmentation is a bit lighter). The fruits in earlier stages of development cannot be differentiated from those of ‘normal’ black elder. The plants (shrubs or small trees) reach normal size, and when there are no leaves and fruits, they cannot be differentiated from other *S. nigra* genotypes [5].

Black elderberry fruits are among the richest sources of flavonoids of the various berry species [6], especially anthocyanins, which give the fruit an extremely dark purple, almost black color. Both anthocyanins and other phenolic substances, e.g., phenolic acids, flavonol glycosides, flavanols, tannins and anthocyanins [4,7,8], in fruits have many organoleptic, nutritional and health effects. Nutritional and health studies have shown that regular consumption of elderberry products has a positive influence on reducing the incidence of cardiovascular and degenerative diseases, cancer and other modern-day diseases, and also has anti-inflammatory effects [9,10]. Elderberry has also recently been of great interest to the food industry, since it is a very rich source of natural anthocyanins, which could replace the use of synthetic colorants. The use of elderberry can be very broad, since its high content of anthocyanins and other polyphenolic substances and vitamins, as well as its high antioxidant activity, allows its use in the food industry as a natural colorant or as an antioxidant [11].

Plants produce anthocyanins to add color to flowers and fruits to attract pollinators and seed dispensers. The synthesis of anthocyanins requires two types of genes, structural genes that are directly involved in the formation of anthocyanins and regulatory genes that control the transcription of the first type of genes [12,13]. A simplified synthetic pathway of anthocyanins in the fruits of berry species, in which various enzymes (e.g., CHS, FHT, DFR, etc.) are involved, is shown in Figure 1 [12,13]. Naturally occurring mutants are also sometimes found in nature. In some acyanic fruit species, it has been confirmed that the expression of the main genes in the anthocyanidin/flavonoid biosynthesis pathway is reduced in white genotypes [14,15,16,17].

Reports of primary and secondary compounds in the green mutant *S. nigra* var. *viridis* are rare, and there have been no studies on the enzymes responsible for phenylpropanoids, the genes involved and their role in fruit maturation. The aim of our study was therefore to analyse sugars, organic acids, polyphenolics, enzymes and some genes responsible for anthocyanin synthesis in two genotypes of *Sambucus nigra* (black-fruited *S. nigra* and green-fruited *S. nigra* var. *viridis*). The fruits were sampled four times during the last 6 weeks of ripening. Since *S. nigra* var. *viridis* is a very attractive and exceptional species of black elder (exceptional in that it is not black) and highly desired among elderberry growers, our goal was to analyse some details of the chemical profile of its fruits to determine its potential value for use as a medicinal plant, in the food industry and for various domestic purposes. The research results on natural acyanic genotypes could lead to new key information on the synthesis of anthocyanins and other polyphenolics in plants.

## 2. Results

We analyzed the three prevalently present sugars in elderberries, i.e., sucrose, glucose and fructose (Appendix A), with glucose and fructose making up the largest part. In *Sambucus nigra* (NIGRA) fruits, these two main sugars accounted for 62% of the total sugars in the first developmental stage and 89% in the last. Interestingly, the proportion of these two sugars was always higher in *S. nigra* var. *viridis* (VIRIDIS) than in NIGRA (87 to 97% of all sugars). In addition, quantitatively, the fruits of VIRIDIS showed a higher total sugar content in the second developmental stage. During the ripening period, the sugar content increased significantly. The highest sugar content was reached in the third developmental stage, when the NIGRA berries contained 59.5 g of total sugar per kilogram and VIRIDIS berries 61.5 g kg^−1^ FW. Thereafter, sugar concentrations decreased slightly (Figure 2).

The most important acids in the elderberries were citric, malic and quinic acids (Appendix A). Their contents accounted for 94% to 97% of all acids analyzed. Fumaric acid, shikimic acid and oxalic acid were present only in lower concentrations. There were no differences in the total acid content in NIGRA fruits between the sampling intervals (Figure 3). However, a comparison of the acid content of green fruits over the ripening period showed a decrease of acids in the last period. When comparing the acidity between the two genotypes, we found that, in the second stage, the acyanic mutant fruits had a significantly higher acidity than the colored fruits, while the dark fruits contained 40% more acids in the last stage.

The phenolic profile between the two genotypes studied was quite similar, except for some polyphenolics that were not identified in the acyanic mutant: catechin, epicatechin, three procyanidin dimers, both trimers and all anthocyanins (Appendix A). The phenolic content varied considerably during maturation. The highest content of total phenolics analyzed (15,930 µg g^−1^ FW) was found in NIGRA fruits during the third developmental stage, mainly due to the drastic increase in the content of anthocyanins, flavonol glycosides and hydroxycinnamic acids compared to the second stage, when the elderberries had only 7203 µg phenolics g^−1^ FW. In the last stage, the phenolic content of NIGRA fruits decreased by 45.3% compared to the third stage (Figure 4).

Comparing the significant differences between the two genotypes in terms of total polyphenolic content, NIGRA fruits had significantly higher total phenolic contents in the second (1.66 times) and third stage (4.55 times) than the green mutant (VIRIDIS). The ratios of the individual phenolic groups also differed between the genotypes in that the NIGRA berries showed 47% flavonol glycosides, 34% anthocyanins, 3% hydroxycinnamic acids and 14% flavanols in the fourth stage. The main phenolic group in VIRIDIS fruits was flavonol glycosides, which accounted for 88% of the total polyphenolics analyzed, the remainder being hydroxycinnamic acids (HCA, 8.2%), while flavanols, dihydrochalcones and flavanones were present only in trace amounts.

The flavonol group was the most abundant in terms of content. The sum of all flavonols was 4126 to 8168 µg g^−1^ FW in the black genotype and 3072 to 12,440 µg g^−1^ FW in the acyanic genotype (Appendix A). Of the flavonol group, quercetin-3-*O*-rutinoside and quercetin-3-*O*-glucoside were the main representatives in both genotypes, accounting for 71 to 88% of all flavonols in NIGRA fruits and 56% to 59% of the flavonols analyzed in VIRIDIS. Interestingly, VIRIDIS fruits showed an extremely high content of isorhamnetin acetyl hexoside 2, from 890 to 3652 µg g^−1^ FW. The unripe fruits of both genotypes had the highest rutin content by some way during the first developmental stage. The highest content of quercetin-3-glucoside in NIGRA fruits was found in the third stage (5739 µg g^−1^ FW) and in VIRIDIS during the first (4745 µg g^−1^ FW) and last stages (4934 µg g^−1^ FW). In total, ten quercetin glycosides, five isorhamnetin derivatives and two kaempferol glycosides were identified (Appendix A). Quercetin derivatives accounted for 80–95% of the total flavonols analyzed in NIGRA fruits, isorhamnetin glycosides 4–18%, and the proportion of kaempferol derivatives was less than 0.7% of total flavonols. Different proportions were found in VIRIDIS fruits, in which the proportion of isorhamnetin glycosides was extremely high compared to NIGRA fruits, with 32–40% of total flavonols, while quercetin glycosides accounted for 58–67% of total flavonols. In the first and last stages, VIRIDIS fruits had a significantly higher content of total flavonols than those of NIGRA. The opposite was the case in the third stage.

No anthocyanins were found in VIRIDIS fruits, but the NIGRA fruits were extremely rich in anthocyanins. Significantly, the highest content was reached in the third stage, at around 6020 µg total anthocyanins per g fruits (Appendix A). In the last period, the anthocyanin content in elderberries decreased by about half. The prevalent anthocyanin was cyanidin-3-*O*-sambubioside, which accounted for 60% of the total anthocyanin content (TAC) in ripe fruits. In second place was cyanidin-3-*O*-sambubiosyl-5-*O*-glucoside, with 16% of the TAC, followed by cyanidin-3,5-*O*-diglucoside and cyanidin-3-*O*-glucoside (7% and 6% of the TAC, respectively).

The fruits of the green mutant were almost free of flavanols, which represented only 0.2 to 0.9% of the total phenolic content. Only one of the procyanidin dimers was found, and no catechin or apicatechin. As a result, the NIGRA fruits showed significantly higher levels of flavanols in all sampling data and reached up to 7 to 21% of the total phenolic compounds found in NIGRA berries.

A large number (12) of hydroxycinnamic acid (HCA) derivatives were found in the elderberries (Appendix A). Caffeoylquinic acids, and chlorogenic and neochlorogenic acid were prevalently present, and accounted for 68–83% of total HCA in NIGRA fruits and 55–66% in VIRIDIS fruits. There were no differences in the total content of caffeoylquinic acids between the green and black genotypes, except for at the third stage, when the black fruits had a 64% higher content than the acyanic genotype (180.7 µg g^−1^ FW). In both genotypes, their content decreased during the ripening period—by 2.4 times in the NIGRA genotype and 4 times in the VIRIDIS genotype—so that upon reaching maturity, the green VIRIDIS fruits contained only 127.48 µg g^−1^ FW of total caffeoylquinic acids. Among other HCA representatives, two hexosides of caffeic acid, three *p*-coumaric acid derivatives and 2 ferulic acid derivatives were analyzed. At the last sampling, the fruits of the acyanic genotype contained 11.64 µg caffeic acid derivatives, 24.81 µg *p*-coumaric acid derivatives and 67.95 µg ferulic acid derivatives per gram of fruit (FW). Their contents in the NIGRA fruits were in most cases fairly similar, with a few exceptions, when the black genotype showed a significantly lower content of ferulic acid derivatives at stages 1 to 3, compared to the acyanic genotype.

Of the dihydrochalcones, only one representative, phloridzin, was found (Appendix A). Its content decreased with the ripening of the elderberry fruits, so that the NIGRA fruits contained 7.5 times less phloridzin at the last stage compared with the first stage, and the VIRIDIS fruits contained 14 times less phloridzin in the same comparison. Comparing the differences in phloridzin content between the two genotypes, NIGRA fruits had 1.5 to 4.2 times more phloridzin than the VIRIDIS genotype.

With respect to polyphenolic compounds, selected enzymes of their biosynthesis were also investigated. This included measurement of the activities of phenylalanine ammonia lyase (PAL), chalcone synthase (CHS)/chalcone isomerase (CHI), flavanone 3β-hydroxylase (FHT) and dihydroflavonol 4-reductase (DFR). Gene expression was measured for anthocyanidin synthase (ANS), the last enzyme in the anthocyanidin pathway, as there is so far no reliable enzyme assay available. The PAL enzyme had the highest activity during the first stage, after which its activity steadily decreased during the ripening period (Figure 5A). In the last stage, PAL activity in NIGRA fruits was 12 times lower, and 11 times lower for the VIRIDIS genotype, than in the first stage. CHS/CHI activity for NIGRA fruits also showed a decrease of 130% from the first to the last sampling period. In contrast, the activity of the CHS/CHI enzymes increased during fruit ripening in the greenish genotype (Figure 5B).

The enzyme activity of FHT was significantly lower in the VIRIDIS genotype during the whole ripening period compared to the NIGRA genotype (Figure 5C) and decreased even further during the season, while FHT activity in NIGRA fruits remained fairly stable. The activity of FHT was thus 2.7 to 6.5 times lower in VIRIDIS fruits during the second and third stages than in the NIGRA genotype. In the last period investigated, it was 110 times lower. Differences in DFR enzyme activity between the two genotypes increased during the season; NIGRA fruits had only ten times higher activity than VIRIDIS fruits in the first stage, but the difference in the third stage was 174 times higher for the NIGRA genotype (Figure 5D). In the last developmental stage, the activity of the DFR enzyme was very low in VIRIDIS fruits. ANS expression showed a similar pattern (Figure 6, Appendix A). In NIGRA, ANS expression strongly increased during fruit development and reached its highest values in ripe fruits. In VIRIDIS, in contrast, the highest expression ratios were observed at the first sampling date, although the expression was still seven times lower than in NIGRA. Thereafter, ANS expression decreased and remained at a steady low level. In ripe NIGRA fruits, ANS expression was more than 4500 times higher than in ripe VIRIDIS fruits.

## 3. Discussion

Acyanic mutants open up new possibilities for analysis of the synthesis of primary and secondary metabolites, especially anthocyanins. This is the first analysis and chemical identification of a large spectrum of secondary and primary metabolites in the greenish, acyanic mutant of *Sambucus nigra*, which is *S. nigra* var *viridis* (VIRIDIS). We were particularly interested in changes in the flavonoid pathway during the ripening of elderberry fruits, and with the altered synthesis of some phenolic compounds in comparison with common black elderberry.

The black colored NIGRA fruits showed significantly higher contents of the total phenolic compounds analyzed on sampling dates 2 and 3, mainly due to the high content of procyanidins, quercetin derivatives and anthocyanins. Previous studies on elderberry have also reported that flavonoids are a major phenolic subclass in elderberry fruits, rutin in particular having a high concentration in fruits [18].

Anthocyanins are pigments that give a red, blue or purple color to fruits. In elderberries, they are mainly concentrated in the epidermis and in the tissue directly under the skin. Elderberry fruits are very dark violet or almost black in color, and their preparations are used in food technology as natural colorants [19]. Compared to other berry species that are also rich in anthocyanins, the concentrations of total anthocyanins in elderberries are extremely high (6020 µg g^−1^ FW); for example, black currant contains 2075 to 3729 µg g^−1^ anthocyanins [20], black mulberry 576 to 4756 µg g^−1^ FW [21] and chokeberry 18.13 mg g^−1^ DW [22]. The green VIRIDIS genotype lacked anthocyanins completely, and flavanols were present only in trace amounts, representing only 0.2 to 0.9% of total phenolic compounds, while they represent 7 to 21% of total phenolics analyzed in NIGRA berries. The flavanol content in the elderberries decreased during the ripening process, as is reported for many other fruits [12,14,23,24].

High concentrations of flavanols (catechin, epicatechin and various procyanidins) cause an astringent taste in unripe fruit and thus offer protection against pathogenic organisms and animals [25]. It can therefore be postulated that VIRIDIS fruits may be significantly less astringent, since they showed lower levels of flavanols than NIGRA fruits throughout the ripening process. This is probably also why we found significantly fewer fruits on VIRIDIS plants at full maturity than on the black genotype; birds had probably taken fruits from the shrubs before us, although the fruits do not have the visual signs of ripeness. Various white strawberry genotypes have also shown significantly lower levels of flavan-3-ols compared to the red fruit types [24,26] and, similarly, white raspberry varieties did not show procyanidin B1 [27]. Xiong et al. [28] reported that tea leaves (*Camellia sinensis* L.) contain significantly lower concentrations of catechin derivatives in their albino phase compared to green shoots.

The biosynthesis of anthocyanins and other phenolic compounds follows three main pathways: the shikimic acid pathway, resulting in the formation of aromatic amino acids; the phenylpropanoid pathway, which converts phenylalanine into hydroxycinnamic acids; and the flavonoid pathway, which uses *p*-coumaroyl-CoA as a precursor and gives rise to the formation of the different flavonoid classes [29]. PAL is thus a key enzyme located at the interface between primary and secondary metabolism [30], and CHS catalyzes the formation of chalcones, which are the first C15 structures and intermediates in the formation of all flavonoid classes. The consecutive action of CHS, CHI, FHT, DFR and ANS results in the formation of anthocyanidins. A further key enzyme in the flavonoid pathway is DFR, as it provides the precursors for the formation of anthocyanidins and flavanols and competes with flavonol synthase (FLS) for the common dihydroflavonol substrates. The substrate specificity of DFR can also have a major impact on the type of anthocyanin formed and, thus, on the color of the tissue [31,32,33]. However, the observed prevalence of cyanidin derivatives in elderberries, which is in line with earlier studies [34], does not point to a special role of the DFR substrate specificity in anthocyanin formation. The presence of a highly active flavonoid 3′-hydroxylase and absence of flavonoid 3′5′-hydroxylase are thus probably the most important factors determining the hydroxylation pattern of the flavonoids in elderberry.

The polyphenol content and the activity of the corresponding enzyme correlated well in the case of NIGRA, but not in the case of VIRIDIS. For PAL, a dramatically higher activity was observed in the first developmental stage of the NIGRA fruits, whereas during the later stages, activities were comparable in the two genotypes. There was thus a strong decrease in PAL activity in the NIGRA fruits, whereas the VIRIDIS fruits kept their originally low level. This was reflected in a decrease in hydroxycinnamic acids during ripening, which is a result of decreased formation on the one hand, and further downstream conversion in the phenylpropanoid and flavonoid pathways on the other. Unripe VIRIDIS fruits thus had the highest total HCA content (786 µg g^−1^ FW), and their content decreased with ripening to 231 µg g^−1^ FW in the last stage. Previous studies have also reported that the enzyme activity of the PAL enzyme decreases as the fruit ripens [35,36], but increases in the peel of the Malay apple during ripening [37]. High differences in HCA values during fruit development have also been reported in white and red strawberries [24] and raspberries [27].

The NIGRA genotype showed significantly higher activity of CHS/CHI enzymes in the first two developmental stages compared with VIRIDIS. During fruit ripening, the black elder genotype showed a significant decrease in the activity of the CHS/CHI enzymes, whilst CHS/CHI activity slowly increased in the VIRIDIS genotype, so that at the end, CHS/CHI activity was comparable in the two genotypes. The activities of FHT and DFR were significantly lower during all sampling dates in the green genotype. As with CHS/CHI activity, FHT activity decreased during fruit ripening, indicating that the early flavonoid pathway is particularly activated at the beginning of fruit ripening. The late enzymes of the flavonoid pathway, DFR and ANS, in contrast, were strongly activated at the end of fruit ripening, which correlates with the high concentrations of anthocyanins and flavonols in the ripe black fruits at sampling time T3. Thus, we did not find a pronounced two-phase activation of the flavonoid pathway in the elderberries as reported for other berries [38]. The lack of activation of the late enzymes of the anthocyanin pathway in the green genotype redirects the flow from anthocyanin and flavanol formation to increased flavonol formation. The highest content of total flavonols was thus found in overripe fruits. The role of DFR in the delicate correlation between anthocyanin and flavonol formation has been addressed previously in many other species, e.g., [33,39,40,41].

The occurrence of acyanic mutants can have multiple causes, which can range from the loss of single enzyme activities to the complete downregulation of the anthocyanin pathway. DFR and FHT inactivity in acyanic mutants have also been reported in bilberry [16], strawberry [34] and bog bilberry [23]. Zorenc et al. [16] claimed that blocked anthocyanin synthesis in white bilberry fruits was due to decreased expression of FGT and ANS genes, as well as too low expression or low activity of the FHT and DFR enzymes.

In addition to polyphenolic compounds, which modulate color and astringency in fruit, changes in sugars and organic acids are further important aspects of fruit ripening. During the ripening process, the total sugar content of the elderberries increased and reached a peak at the third sampling date. An increase in sugar concentration during ripening has also been reported in other fruit species [15,42]. However, in the last stage, the total sugar content in NIGRA berries decreased by 41% compared to the previous stage, and by 15% in the VIRIDIS genotype. Such a decrease in the total sugar content in the last ripening phase has also been reported in bilberries [43]. This reduction in sugar content from T3 to T4 may be to stop their synthesis and remobilization.

A significant difference in the total sugar content between the two genotypes was found only during the second stage, when acyanic berries had a content 2.17 times higher than that of NIGRA berries. This difference between the two genotypes was due to the higher glucose and fructose content of the VIRIDIS genotype in the second stage. This is probably related to the synthesis of anthocyanins in NIGRA, where sugars are spent for their synthesis, so that the sugars remain in higher concentrations in the fruits of the green mutant. Guan et al. [44] also reported that the white grape variety (‘Gamay’) had a 28% higher fructose and glucose content than the red mutant of the same grape variety (‘Gamay Freaux’). The black genotype NIGRA, however, had a significantly higher sucrose content than the green genotype VIRIDIS (2.1 to 3.2 times higher) in all periods studied.

Organic acids in fruits, which can be present in high concentrations, also have an influence on the fruits’ taste and degree of ripeness. The concentrations of total organic acids remained almost unchanged during fruit development in the NIGRA genotype, while VIRIDIS fruits showed significantly lower acidity levels upon maturation compared to the previous stages. This is in contrast to other berries, for which a decrease in acid content in the fruits during ripening has been reported in previous studies [45,46]. The most important acids in elderberries were quinic, malic and citric acid. A significant difference between the investigated genotypes occurred only during the second stage, when NIGRA fruits had a 1.3 times lower acid content, and in the last, when they had a 1.4 times higher content than VIRIDIS fruits. Nishad et al. [47] previously reported that white pomelo genotypes had higher acid concentrations than pink and red genotypes.

## 4. Materials and Methods

### 4.1. Plant Material

The investigation involved two *S. nigra* genotypes with phenotypically very different fruits: (1) common local black elderberry with dark purplish black fruits, and (2) the botanical variety *viridis,* characterized by light whitish green fruits. The shrubs were 17 years old at the time of sampling (planted in 2002), growing in the botanical garden of the University of Maribor (latitude: 46°30′ N, longitude: 15°38′ E), at Pivola, in the southern suburbs of the city of Maribor, Slovenia. The plants were growing in optimal environmental conditions, fully exposed to sun.

The fruit samples of black berries (*Sambucus nigra* var. *nigra* abrev NIGRA) and natural white mutant green berries (*S. nigra* var. *viridis* abrev VIRIDIS) were collected during the ripening period of 2019, in four characteristic developmental stages of the fruit. The sampling dates were 30 June (T1—only a few berries of the NIGRA clusters were red), 15 July (T2—approx. 60% of berries in a cluster were red-violet colored), 29 July (T3—when all individual berries had reached full ripeness (berries had reached their maximum size and final color) (Figure 7) and 12 August (T4—when berries became softer (over-ripe stage)). We also measured the dry matter at each sampling date. On the first three dates, the DM for NIGRA berries was 18.2–19.5%, and for VIRIDIS 18.1–19.7%. On the last sampling date, when the berries were overripe, the dry matter was the highest, at 22.8% for NIGRA and 22.5% for VIRIDIS. After collecting the samples in the botanical garden, the fruits were quickly transported in a cooling bag to the laboratory, where they were immediately frozen in liquid nitrogen and then stored at −80 °C. For the analysis of all chemical components of elderberry, 4 separate biological replicates were prepared on each sampling date. On the same biological replicate, analyses of sugars, acids, and phenolic substances and measurement of enzyme activity were performed in four replicates.

### 4.2. Chemicals

For the determination of sugars and organic acids, we used the following standards: sucrose, glucose, fructose, tartaric, quinic, malic, citric, oxalic, shikimic and fumaric acid from Sigma-Aldrich (Steinheim, Germany). For phenolic compounds’ identification and calculation, the following standards were purchased: procyanidin B1, *p*-coumaric acid, epicatechin ferulic acid, catechin, quercetin-3-*O*-rutinoside, naringenin, quercetin-3-*O*-galactoside, quercetin-3-*O*-glucoside, quercetin-3-*O*-xyloside, kaempferol-3-*O*-glucoside and phloridzin from Fluka Chemie (Seelze, Germany), quercetin-3-*O*-arabinofuranoside from Apin Chemicals LTD (Compton, UK), isorhamnetin-3-*O*-glucoside from Extrasynthèse (Genay, France), and chlorogenic and caffeic acid, quercetin-3-*O*-rhamnoside, 3-*O*-caffeoylquinic acid, cyanidin-3-*O*-glucoside, pelargonidin-3-glucoside, cyanidin-3-*O*-rutinoside and 4-*O*-caffeoylquinic acid from Sigma-Aldrich. For the extraction of phenolic compounds, methanol from Sigma-Aldrich, double-distilled water purified with a Milli-Q system (Millipore, Bedford, MA, USA) and 3% formic acid from Fluka Chemie were used. The chemicals for the mobile phases were acetonitrile (HPLC-MS), formic acid, sulphuric acid and double-distilled water.

### 4.3. Extraction and Determination of Sugars and Organic Acids

Sugars and acids were extracted using the method reported by Mikulic-Petkovsek et al. [48]. The elderberries were ground into powder with liquid nitrogen, and 500 mg of the sample was weighed and made up with 2.5 mL double-distilled water. For each genotype (black and green), four replicates of several fruits were performed at each sampling date (*n* = 4). The extraction of primary metabolites from the berries took half an hour at room temperature, with continuous shaking. The samples were then centrifuged at 15.695× *g* (Eppendorf Centrifuge 5819 R, Eppendorf, Hamburg, Germany). The supernatant was filtered through cellulose ester (Macherey-Nagel, Düren, Germany) filters into a vial.

The chromatographic conditions of the analysis are described in Mikulic-Petkovsek et al. [48]. For the analysis of organic acids, a UV detector set at 210 nm, mobile phase 4 mM H_2_SO_4_ and a Rezex ROA column, organic acid H^+^ (8%) (300 mm × 7.8 mm, Phenomenex) heated to 65 °C were used. For the analysis of sugars, an RI detector was used; the mobile phase was distilled water and a Rezex RCM monosaccharide Ca^+^ (2%) column (300 mm × 7.8 mm; Phenomenex, Torrance, CA, USA) heated to 65 °C. For each of the two analyses, the total run time was 30 min.

Sugars and acids in elderberries were identified based on their retention time in the chromatogram and by adding external standards to samples. Their content was calculated from a calibration curve of the external standards and expressed in g kg^−1^ fresh weight.

### 4.4. Extraction and Determination of Individual Phenolic Compounds

The extraction and identification of phenolic compounds in elderberries was performed as described by Mikulic-Petkovsek et al. [48]. The fruits were ground to powder with liquid nitrogen; 500 mg of the material was then weighed, and 2.5 mL 80% methanol with 3% formic acid poured over. Extraction was performed in an ice-cold ultrasonic bath for 50 min. The extracts were then centrifuged at 15.695× *g* and filtered through 0.20 µm polyamide filters (chromafil AO-20/25, Macherey-Nagel) into vials. Phenolic compounds were analyzed by HPLC (Dionex, Sunnyvale, CA, USA) using a PDA detector and scanning at three wavelengths: 530 nm for anthocyanins, 350 nm for flavonol glycosides and 280 nm for other phenolic compounds. The analytical conditions were the same as those previously reported by Mikulic-Petkovsek et al. [48]. Mobile phase A was 96.9% H_2_O/3% acetonitrile/0.1% formic acid, and mobile phase B was 96.9% acetonitrile/3% water/0.1% formic acid. A Gemini C18 column heated to 25 °C (150 × 4.6 mm, 3 μm; Phenomenex) was used to determine the phenolic substances. The analysis time was 45 min and the gradient was summarized using the method reported by Mikulic-Petkovsek et al. [49].

Identification of phenolic compounds in the elderberries was performed with an LTQ XLTM Linear Ion Trap Mass Spectrometer (Thermo Scientific, Waltham, MA, USA) with electrospray ionization (ESI), working in positive (for anthocyanins) and negative (for other phenolic groups) ion mode and scanning from *m*/*z* 110–1600. Phenolic compounds were identified based on a comparison of fragmentation patterns with literature information, the UV spectral characteristics of each compound and peaking with standards. The quantification of the phenolic compounds was performed using calibration curves. The content of phenolic compounds was expressed in µg g^−1^ FW elderberry.

### 4.5. Extraction and Determination of Enzyme Activities

The activity of the following enzymes was analyzed: phenylalanine ammonia-lyase (PAL; EC 4.3.1.24), chalcone synthase (CHS; EC 3.2.1.74)/chalcone isomerase (CHI; EC3.2.1.14), flavanone 3β-hydroxylase (FHT; EC 1.14.11.9) and dihydroflavonol 4-reductase (DFR; EC 1.1.1.219), the key enzymes of the phenyl-propanoid pathway. Enzyme activities were measured according to the protocol described by Halbwirth et al. [50], with some modifications. Shock-frozen elderberries were ground to powder with liquid nitrogen, and 400 mg berry powder was homogenized in a mortar with 200 mg quartz sand (Sigma-Aldrich); to disrupt the fruit cells, 200 mg polyclar AT and 3 mL Dellus buffer, which contained 0.1 M HEPES (4-(2-hydroxyethyl)-1-piperazineethanesulfonic acid), 40 mM sucrose, 0.75 mM polyethylene glycol 20,000, 0.1 M sodium ascorbate, 1 mM dithioerythritol, and 0.025 mM CaCl_2_ (pH 7.3) were used. The sample homogenates were centrifuged for 15 min at 12,257× *g* at 4 °C. To remove the low-molecular components, 400 μL supernatant was passed over a gel chromatography column (Sephadex G25 medium) Sigma-Aldrich (Steinheim, Germany). The solution of proteins eluted in the excluded volume of the column (crude extract) was used for enzyme assays. Enzyme activity was determined as described by Halbwirth et al. [38] under conditions optimized for elderberry fruits. Detailed assays for enzymes PAL, CHS/CHI, FHT and DFR are presented in Appendix A. The samples were heated at 30 °C for 20 min. To quantify the protein content, the modified Lowry method [51] was used to determine the specific enzyme activity. The activity for all enzymes (PAL, FHT, DFR and CHS/CHI) was calculated and expressed as nkat kg^−1^ total protein.

### 4.6. Quantitative Real-Time PCR

mRNA was extracted from elderberries with a μMACS mRNA isolation kit (Miltenyi Biotec, Bergisch Gladbach, Germany). Using the primer oligo-dT SMART (AAGCAGTGGTATCAACGCA GAGTAC(T23)VN), cDNA was synthesized by SuperScript II Reverse Transcriptase (Invitrogen, Waltham, MA, USA). The coding sequence of the *ANS* gene (NCBI OP150920) was extracted from a *Sambucus canadinensis* transcriptome study (PRJEB21674). The deduced amino acid sequence showed 78% identity compared to the ANS of *Arabidopsis thaliana* (NCBI Q96323), and 98% to the ANS of *Lonicera caerulea* (NCBI ALU09330). *ANS* gene expression was evaluated by qPCR using the StepOnePlus system (Applied Biosystems, Hamburg, Germany) and SybrGreenPCR Master Mix (Applied Biosystems, Vienna, Austria) according to the manufacturer’s protocol, using the qPCR primer pairs (Appendix A). Product specificity was confirmed by analysis of melting curves and gel electrophoresis. The analysis was performed in four independent replicates, and the results were normalized to elongation factor 2 as the reference gene. The relative expression ratio was calculated according to Pfaffl [52].

### 4.7. Statistical Analysis

The differences between the two elderberry gentoypes (black and acyanic type) as well as the differences between the developmental stage sampling dates in the content of sugars, organic acids, phenolic compounds and enzyme activities were tested using the one-way ANOVA method. The differences between the genotypes were tested with the LSD test, and the differences between the sampling data with the Duncan test at a significance level of 0.05. The statistical program Statgraphics Centurion was used for the statistical analysis. 

## 5. Conclusions

Our work is the first in-depth evaluation of a long-known rare genotype of *S. nigra*, which is a green mutant of black elderberry. In both genotypes, a broad range of metabolites, namely polyphenolics, sugars and organic acids, were analyzed at four stages of fruit ripening. In addition, the underlying pathway leading to the formation of anthocyanins was investigated by measuring the enzyme activities of PAL, CHS/CHI, FHT and DFR, as well as the ANS gene expression. The differences in the chemical composition between black elderberry fruits and the greenish mutant are due to genetic variability. No anthocyanins were found in the green genotype, which can be explained by the lack of activation of the late enzymes of the flavonoid pathway in the late stages of berry development. However, a high content of sugars and flavonols was found instead. The black genotype had the typical high anthocyanin content of elderberries, which seems to be a result of a strong induction of the late flavonoid pathway in the late berry developmental stages. The results of our research will contribute to elucidation of the regulatory structures of the flavonoid pathway and anthocyanin synthesis in fruit. Furthermore, such mutants can be used to investigate the contribution of distinct secondary metabolites to observed beneficial health effects.

## Figures and Tables

**Figure 1 plants-12-00504-f001:**
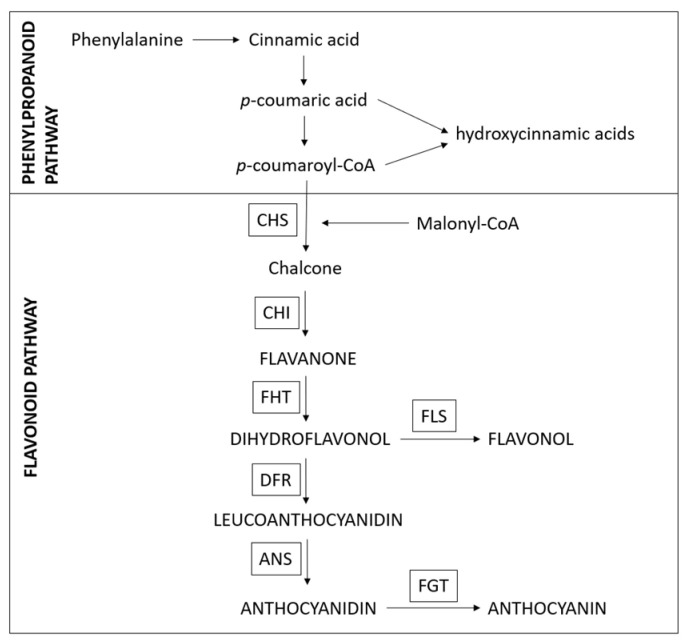
Simplified synthetic pathway of anthocyanins in the fruits of berry species. PAL—phenylalanine ammonia lyase; CHS—chalcone synthase; CHI—chalcone isomerase; FHT—flavanone 3-hydroxylase; DFR, dihydroflavonol 4-reductase; ANS—anthocyanidin synthase; FLS—flavonol synthase; FGT—flavonoid 3-*O*-glucosyltransferase [12,13].

**Figure 2 plants-12-00504-f002:**
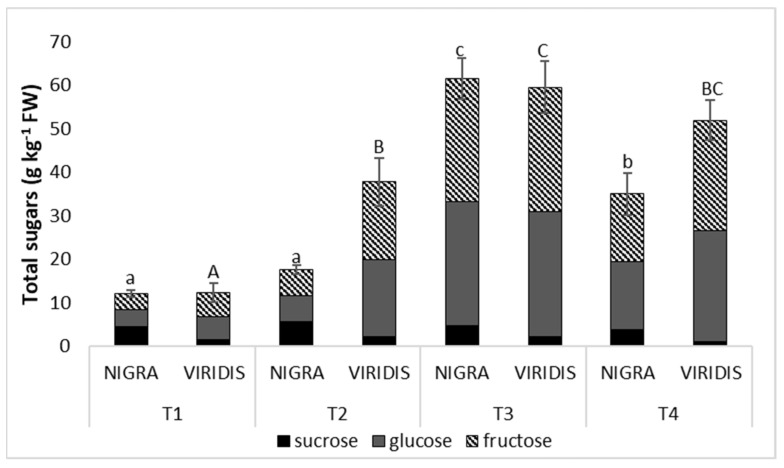
Content of each sugar and total sugars (g kg^−1^ FW) of *Sambucus nigra* (NIGRA) and *S. nigra* var. *viridis* (VIRIDIS) fruits during different developmental stages (T1–T4). Different small letters (a–c) denote statistically significant differences between developmental stages for the NIGRA genotype, and different capital letters (A–C) denote statistically significant differences between developmental stages for the VIRIDIS genotype at *p* < 0.05 (Duncan test).

**Figure 3 plants-12-00504-f003:**
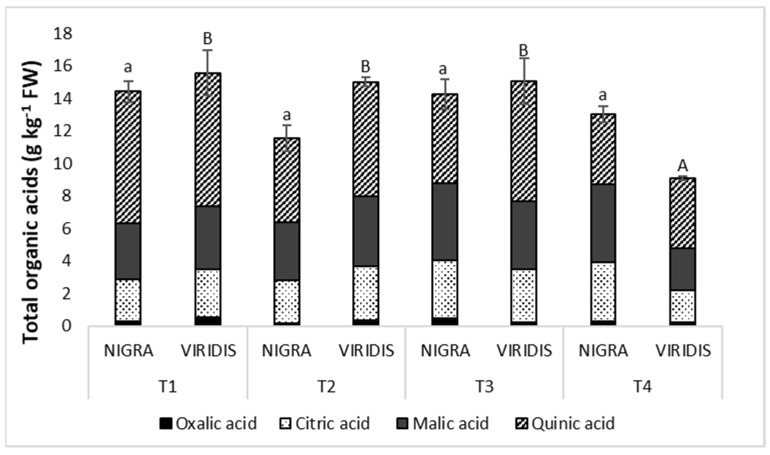
Content of each organic acid and total acids (g kg^−1^ FW) of *Sambucus nigra* (NIGRA) and *S. nigra* var. *viridis* (VIRIDIS) fruits during different developmental stages (T1–T4). Different small letters (a–b) denote statistically significant differences between developmental stages for the NIGRA genotype, and different capital letters (A–B) denote statistically significant differences between developmental stages for the VIRIDIS genotype at *p* < 0.05 (Duncan test).

**Figure 4 plants-12-00504-f004:**
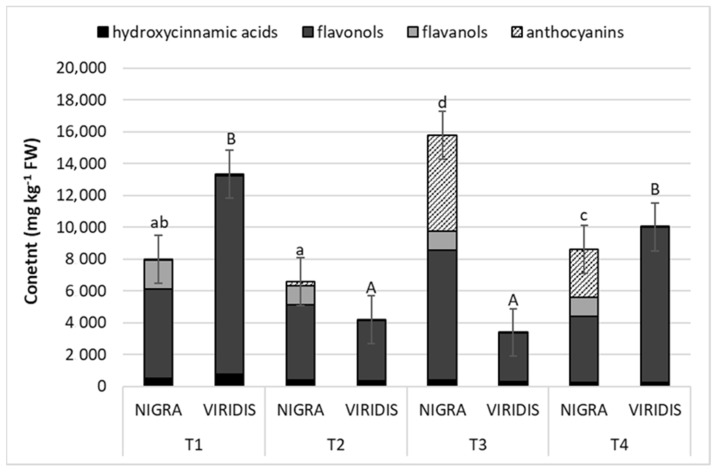
Content of the main polyphenolic classes (g kg^−1^ FW) of *Sambucus nigra* (NIGRA) and *S. nigra* var. *viridis* (VIRIDIS) fruits during different developmental stages (T1–T4). Different small letters (a–d) denote statistically significant differences between developmental stages for the NIGRA genotype, and different capital letters (A–B) denote statistically significant differences between developmental stages for the VIRIDIS genotype at *p* < 0.05 (Duncan test).

**Figure 5 plants-12-00504-f005:**
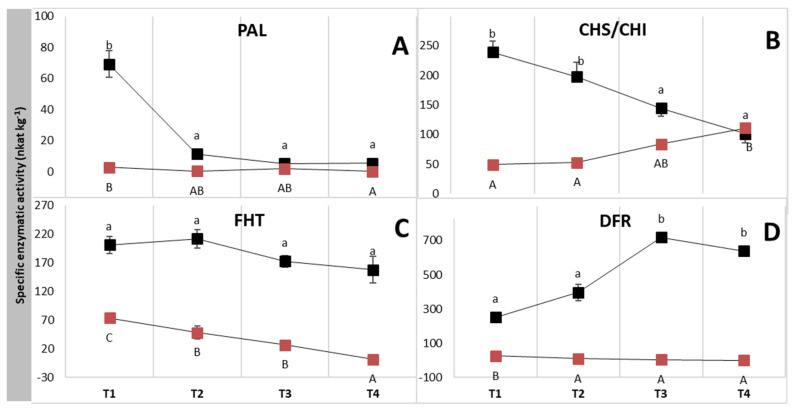
Specific enzymatic activities of PAL—phenylalanine ammonia-lyase (**A**), CHS/CHI—chalcone synthase/chalcone isomerase (**B**), FHT—flavanone 3β-hydroxylase (**C**) and DFR—dihydroflavonol-4-reductase (**D**) (nkat kg^−1^) in NIGRA (■) and VIRIDIS (■) berries during different developmental stages (T1–T4). Different small letters (a–b) denote statistically significant differences between developmental stages for the NIGRA genotype, and different capital letters (A–C) denote statistically significant differences between developmental stages for the VIRIDIS genotype obtained with the Duncan test (*p* < 0.05). Data are presented as average value ± standard error (*n* = 4).

**Figure 6 plants-12-00504-f006:**
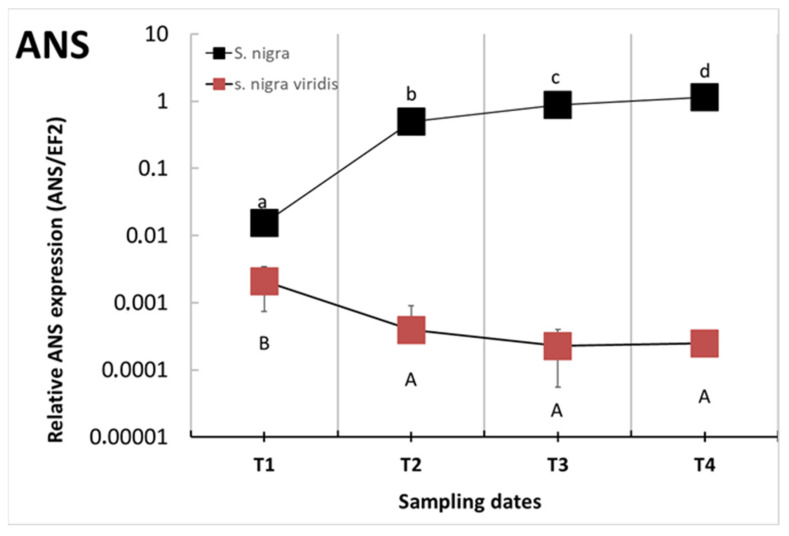
Expression level of ANS in NIGRA (■) and VIRIDIS (■) berries during different developmental stages (T1–T4). Quantitative expression was normalized to *elongation factor 2*. The *y*-axis shows a logarithmic scale to better visualize the differences between NIGRA and VIRIDIS. A non-logarithmic version is provided in Appendix A, Different small letters (a–d) denote statistically significant differences between developmental stages for the NIGRA genotype, and different capital letters (A–B) denote statistically significant differences between developmental stages for the VIRIDIS genotype, obtained with the Duncan test (*p* < 0.05). Data are presented as average value ± standard error (*n* = 4).

**Figure 7 plants-12-00504-f007:**
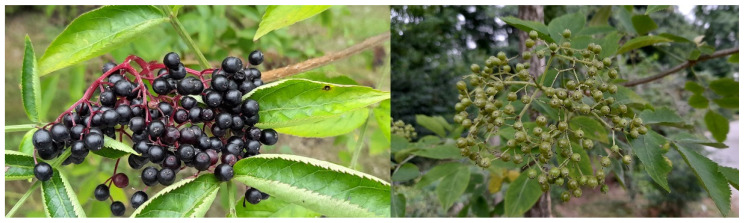
Plant material for the study. On the left picture are fruits of *Sambucus nigra* var. *nigra,* and on the right picture are fruits of *Sambucus nigra* var. *viridis* on the third sampling date (29 July).

## Data Availability

The data presented in this study are available in the article and the Appendix A.

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
