# Peer review of "Biochemical Characterization of Black and Green Mutant Elderberry during Fruit Ripening"

_plants, 2023, doi:10.3390/plants12030504_

Round 1

Reviewer 1 Report

Keywords and Abstract are well addressed. The Introduction is well structured, clear and well aiming. Results are well supported by adequate statistical analysis. Discussions are of good quality, well supported by appropriate references. Materials and methods are well structured and written, the used methods being appropriate and briefly described. It discussed appropriately the obtained findings. The used references are relevant and up-to-date.

Author Response

Dear reviewer.

Thank you for your positive comments on the article.

The authors.

Reviewer 2 Report

The study is informative and meaningful. However, the manuscript needs to be improved for publication. 

1. Please use SI units, keep the consistency in using the units, and be careful for expression of units. For example,  ug/g FW  ugg-1 FW, 15903 15,903, rpm → gravity unit

2. Needs spell-check and proofread Englsih writing. For example, line 16: selected enzymes of → enzyme activities selected in, line 86: acynic → acyanic, line 214: date → stage, line 212; at the 3 first sampling → at 1 to 3 stages; line 226: 11 times? and similar → different?, line 246: not active → very low, line 331: decreased →increased, line 374 and 379: acidity acid contents or acids, etc.

3. Be careful in using abbreviation. after indicating abbreviation, using abbreviation is required and full spell is not allowed.

4.  The graph should be drawn with lines for x- and y-axis and legends should be moved into inside. Please modify the expression of the unit and put the names of x-axis, Cultivars and ripening stages. In comparisons of mean values, use a in the highest, b in the second one, c in the third one, etc. and compare the significance among all stages and cultivars, together. Also, the comparisons among individual component are recommended, too. 

 5. Improve some confusion in some parts. For example, in line 315, authors mentioned correlated well, but it was not true because CHS/CHI was not correlated well. Also, PAL was higher in NIGRA, but the content of hydroxycinnamic acids was low. Why? It needs discussion. In line 340-344, depending on stages, enzyme activities were not matched well. Then, modify these sentences. In line 365, I couldn’t agree it because total phenolics content was higher in VIRIDS. Why are you mentioned just anthocyanin, not phenolics. Also, the sentence of line 430-431 need for modification. 7 replicates here or 4 replicates in Figures?

6. Where is Table 1. Please check line 173. 

7. Please add the composition of extraction buffer in Materials and Methods.

8. Where is the data of dry matter? Why dis you mentioned it in line 98-401.

Author Response

Reviewer’s comments:

The study is informative and meaningful. However, the manuscript needs to be improved for publication. 

We have accepted all the reviewer's comments and included them in the text of the article. We are very grateful to the reviewer for all the precise corrections and comments that improved the quality of the manuscript.

  1. Please use SI units, keep the consistency in using the units, and be careful for expression of units. For example,  ug/g FW →ug∙g-1 FW, 15903 → 15,903, rpm → gravity unit

We have changed all units to SI units.

  1. Needs spell-check and proofread English writing. For example, line 16: selected enzymes of → enzyme activities selected in, line 86: acynic → acyanic, line 214: date → stage, line 212; at the 3 first sampling → at 1 to 3 stages; line 226: 11 times? and similar → different?, line 246: not active → very low, line 331: decreased →increased, line 374 and 379: acidity → acid contents or acids, etc.

All suggested corrections have been made in the manuscript.

  1. Be careful in using abbreviation. after indicating abbreviation, using abbreviation is required and full spell is not allowed.

We have made all corrections in the manuscript.

4.The graph should be drawn with lines for x- and y-axis and legends should be moved into inside. Please modify the expression of the unit and put the names of x-axis, Cultivars and ripening stages. In comparisons of mean values, use a in the highest, b in the second one, c in the third one, etc. and compare the significance among all stages and cultivars, together. Also, the comparisons among individual component are recommended, too. 

Thank you very much for the relevant comment. Based on the reviewer's comment, we have corrected the graph. In labeling the statistical differences, we used the letters provided by the statistical program R commander so we did not change them. The first letter of the alphabet always represents the lowest mean value. We could also mark according to the reviewer's suggestion, but this would not improve anything. The statistical significance remains the same.

  1. Improve some confusion in some parts. For example, in line 315, authors mentioned correlated well, but it was not true because CHS/CHI was not correlated well. Also, PAL was higher in NIGRA, but the content of hydroxycinnamic acids was low. Why? It needs discussion. In line 340-344, depending on stages, enzyme activities were not matched well. Then, modify these sentences. In line 365, I couldn’t agree it because total phenolics content was higher in VIRIDS. Why are you mentioned just anthocyanin, not phenolics. Also, the sentence of line 430-431 need for modification.

We thank the reviewer for his comments, which were very helpful. Based on his comments, we have corrected some unclear sentences in the discussion

The polyphenol content and the activity of the corresponding enzyme correlated well in the case of NIGRA but not in the case of VIRIDIS. The correlation between enzyme activity and phenolic content is often not good. The accumulation of phenolic substances depends not only on the activity of certain synthesis enzymes but may also be the result of decomposition processes, which are more frequent in the later stages of fruit ripening. For PAL, a dramatically higher activity was observed in the first developmental stage of the NIGRA fruits, whereas during the later stages, activities were comparable in the two genotypes. There was thus a strong decrease of PAL activity in the NIGRA fruits, whereas the VIRIDIS fruits kept their original low level. This was reflected in a decrease of hydroxycinnamic acids during ripening, which is a result of decreased formation, on the one hand, and further downstream conversion in the phenylpropanoid and flavonoid pathways on the other. The content of hydroxycinnamic acids in NIGRA was low because they are further converted to other flavonoids and do not accumulate as in VIRIDIS, due to the lower enzyme activities downstream.

Reviewer comment: In line 340-344, depending on stages, enzyme activities were not matched well. Then, modify these sentences.

The late enzymes of the flavonoid pathway, DFR and ANS, in contrast, were strongly activated at the end of fruit ripening, which correlates with the high concentrations of anthocyanins and flavonols in the ripe black fruits at sampling time T3. Thus, we did not find a pronounced two-phase activation of the flavonoid pathway in the elderberries, as reported for other berries [38]. The lack of activation of the late enzymes of the anthocyanin pathway in the green genotype redirects the flow from anthocyanin and flavanol formation to increased flavonol formation. The highest content of total flavonols were thus found in overripe fruits. The role of DFR in the delicate correlation between anthocyanin and flavonol formation has been addressed previously in many other species, e.g., [33,39,40,41].

Reviewer comment: In line 365, I couldn’t agree it because total phenolics content was higher in VIRIDS. Why are you mentioned just anthocyanin, not phenolics.

After reexamining the results, we found the following: NIGRA has a statistically higher content of total phenolics analyzed in the second and third terms compared to VIRIDIS, while there are no significant differences between genotypes on the first and fourth dates. The higher content of total phenolics analyzed is mainly due to the higher content of anthocyanins and flavonols in T2 and T3. We have supplemented the discussion accordingly.

  1. Where is Table 1. Please check line 173. 

It was a mistake, SupplementaryTable S3 is correct.

  1. Please add the composition of extraction buffer in Materials and Methods.

We have added additional text in the Materials and Methods.

  1. Where is the data of dry matter? Why did you mentioned it in line 98-401.

The data on the percentage of dry matter were given only in Material and Methods and were included as additional information.

The authors

Reviewer 3 Report

Very interesting paper, although sampled plant material was represented only by two individual elderberry plants grown in "almost" same environment conditions...

Sampling more individuals should be taken into consideration in the future, either clonally propagate these two genotypes and set up a trial with replications or to collect samples of the two fenotypes from different locations and carry on analysis at least two years. Results might be interesting too taking into account the variability of the soil and the variability of climatic conditions into different years.

Author Response

Dear Reviewer.

Thank you for your positive opinion about the article and your comment for our further research. We agree with you that it is always better to perform analyzes on a larger number of samples. However, since we were limited by the plant material, we performed only four replicates per individual genotype and individual sampling. We believe that it would also be interesting to test the content of components in plants growing in different locations and in different years, but this was not the subject of our research. The main objective of the study was to investigate how the profile of primary and secondary metabolites differs in two different elderberry genotypes. We will consider your suggestion in the future. We thank you.

The authors.